# 9-Methylfascaplysin Is a More Potent Aβ Aggregation Inhibitor than the Marine-Derived Alkaloid, Fascaplysin, and Produces Nanomolar Neuroprotective Effects in SH-SY5Y Cells

**DOI:** 10.3390/md17020121

**Published:** 2019-02-18

**Authors:** Qingmei Sun, Fufeng Liu, Jingcheng Sang, Miaoman Lin, Jiale Ma, Xiao Xiao, Sicheng Yan, C. Benjamin Naman, Ning Wang, Shan He, Xiaojun Yan, Wei Cui, Hongze Liang

**Affiliations:** 1Ningbo Key Laboratory of Behavioral Neuroscience, Zhejiang Provincial Key Laboratory of Pathophysiology, School of Medicine, Ningbo University, Ningbo 315211, China; 15658223676@163.com (Q.S.); xx15058492711@163.com (X.X.); Yansicheng9@163.com (S.Y.); 2Li Dak Sum Yip Yio Chin Kenneth Li Marine Biopharmaceutical Research Center, College of Food and Pharmaceutical Sciences, Ningbo University, Ningbo 315211, China; bnaman@nbu.edu.cn (C.B.N.); wangning2@nbu.edu.cn (N.W.); heshan@nbu.edu.cn (S.H.); yanxiaojun@nbu.edu.cn (X.Y.); 3Key Laboratory of Industrial Fermentation Microbiology of Education, State Key Laboratory of Food Nutrition and Safety, College of Biotechnology, Tianjin University of Science & Technology, Tianjin 300457, China; fufengliu@tust.edu.cn (F.L.); bixian2b@163.com (J.S.); 4School of Materials Science and Chemical Engineering, Ningbo University, Ningbo 315211, China; lmm_dll@163.com (M.L.); mjl2137297289@163.com (J.M.)

**Keywords:** fascaplysin, Alzheimer’s disease, Aβ, oligomer, β-carboline

## Abstract

β-Amyloid (Aβ) is regarded as an important pathogenic target for Alzheimer’s disease (AD), the most prevalent neurodegenerative disease. Aβ can assemble into oligomers and fibrils, and produce neurotoxicity. Therefore, Aβ aggregation inhibitors may have anti-AD therapeutic efficacies. It was found, here, that the marine-derived alkaloid, fascaplysin, inhibits Aβ fibrillization *in vitro*. Moreover, the new analogue, 9-methylfascaplysin, was designed and synthesized from 5-methyltryptamine. Interestingly, 9-methylfascaplysin is a more potent inhibitor of Aβ fibril formation than fascaplysin. Incubation of 9-methylfascaplysin with Aβ directly reduced Aβ oligomer formation. Molecular dynamics simulations revealed that 9-methylfascaplysin might interact with negatively charged residues of Aβ_42_ with polar binding energy. Hydrogen bonds and π–π interactions between the key amino acid residues of Aβ_42_ and 9-methylfascaplysin were also suggested. Most importantly, compared with the typical Aβ oligomer, Aβ modified by nanomolar 9-methylfascaplysin produced less neuronal toxicity in SH-SY5Y cells. 9-Methylfascaplysin appears to be one of the most potent marine-derived compounds that produces anti-Aβ neuroprotective effects. Given previous reports that fascaplysin inhibits acetylcholinesterase and induces P-glycoprotein, the current study results suggest that fascaplysin derivatives can be developed as novel anti-AD drugs that possibly act via inhibition of Aβ aggregation along with other target mechanisms.

## 1. Introduction

Alzheimer’s disease (AD) is a chronic and debilitating neurodegenerative disease that normally occurs in the elderly. AD is clinically characterized by cognitive impairments, mainly due to the loss of synapses and neurons in the brain, representing the greatest of health challenges [1]. It is projected that more than 100 million people may be affected by AD by 2050 [2]. Unfortunately, no effective drug is currently available to cure or prevent this disease. Given the huge socioeconomic burden associated with AD on global healthcare and medical systems, there is an urgent need for the development of new drugs to treat and/or prevent AD [3].

Two features of AD are the accumulation of extracellular senile plaques and intracellular neurofibrillary tangles. The amyloid hypothesis is now recognized as the core pathology of AD. This hypothesis explained that β-amyloid (Aβ), the main component of senile plaque, is a chief pathological element of dementia [4]. Aβ itself normally consists of 38–42 amino acids, and the initial Aβ monomer can aggregate into fibrils and toxic soluble oligomers [5]. Previous studies have shown that plaque deposit levels do not necessarily equate to the severity of AD in patients. Soluble Aβ oligomer has been determined to be much more toxic than mature fibrils or plaques, and is widely regarded as the main neurotoxin in the progress of AD [6]. The inhibition of Aβ aggregates, and soluble Aβ oligomer in particular, therefore, might be a therapeutic mechanism for new drugs to be developed to treat AD. Many in vitro protocols exist that could simulate Aβ aggregation in vivo, and these produce Aβ oligomers and/or fibrils, leading to the ease of studying Aβ aggregation inhibitors.

The ocean is a valuable resource for the discovery of new biologically active natural products [7]. Marine organisms, in particular sponges, can overcome drastic environmental changes and produce a variety of secondary metabolites. Compounds produced by marine sponges have been reported with many bioactivities, such as antioxidant, anti-inflammatory, antitumor, antidiabetic, and antibacterial properties [8,9,10]. Fascaplysin is a widely active benzoyl-linked β-carboline alkaloid that was originally isolated from the Fijian sponge *Fascaplysinopsis* sp. [11]. Fascaplysin has been previously demonstrated to have anticancer and antioxidative stress properties [11]. Manda et al. later reported that fascaplysin inhibits the activity of acetylcholinesterase, and induced P-glycoprotein expression and activity, suggesting that this compound might produce neuroprotective effects [12]. With the aim of enhancing the activity of fascaplysin on acetylcholinesterase, fascaplysin derivatives were designed, including 9-methylfascaplysin in particular. Based on molecular modeling studies, it was suggested that 9-methylfascaplysin would be the best analogue designed to occupy the binding pocket of acetylcholinesterase. Therefore, 9-methylfascaplysin was synthesized for biological interrogation. The present study has shown that fascaplysin can directly inhibit Aβ fibrillization in vitro. 9-Methylfascaplysin more potently inhibited Aβ fibrillization than fascaplysin. Moreover, 9-methylfascaplysin directly interacts with Aβ, inhibiting the formation of Aβ oligomers and preventing Aβ oligomer-induced neuronal death at nanomolar concentrations. Finally, molecular dynamics simulations suggested some specific interactions between 9-methylfascaplysin and the negatively charged residues of Aβ.

## 2. Results

### 2.1. The New Fascaplysin Analogue, 9-Methylfascaplysin, Is a More Potent Aβ Fibrilization Inhibitor

Fascaplysin (**3a**) and 9-methylfascaplysin (**3b**) were synthesized by a two-step process from commercially available tryptamine and 5-methyltryptamine, respectively, as shown in Figure 1. This represents the first report of 9-methylfascaplysin, as this molecule was newly designed and synthesized for this study. The nuclear magnetic resonance spectra of ^1^H NMR and ^13^C NMR, and HRMS of fascaplysin and 9-methylfascaplysin, were shown in Appendix A.

Aβ fibrils are considered to be neurotoxic and have been correlated with the progress of AD in previous studies [13]. Therefore, it was explored whether fascaplysin and 9-methylfascaplysin could affect the formation of Aβ fibrils. Thioflavin-T (ThT) can specifically bind to fibrillar structures and produce a shift in the emission spectrum, allowing quantitation because the fluorescent signal is proportional to the amount of fibril [14]. Furthermore, it is known that Aβ forms fibrils after incubation at 37 °C in the dark. By using this in vitro system, it was found that fascaplysin (3–10 μM) slightly, but statistically significantly, inhibits the formation of Aβ fibrils. Moreover, 9-methylfascaplysin (0.3–30 μM) significantly inhibited Aβ fibrilization in a concentration-dependent manner, showing that this compound is more powerful than fascaplysin in reducing the amounts of Aβ fibrils (Figure 2).

### 2.2. The Formation of Aβ Oligomer is Inhibited by 9-Methylfascaplysin

Aβ oligomer is the most toxic species in amyloid pathogenesis. The inhibition of Aβ oligomer by small molecules is regarded as a promising disease-modifying strategy for the treatment of AD. To evaluate whether 9-methylfascaplysin could inhibit Aβ oligomerization, dot blotting analysis and TEM were used. It is reported that Aβ forms oligomers by continuous shaking in vitro [15]. Using this protocol, Aβ oligomer samples were prepared, and 9-methylfascaplysin was tested for its ability to modified Aβ. It was demonstrated that co-incubation of 9-methylfascaplysin with Aβ effectively decreased the formation of Aβ oligomer as shown by the dot blotting analysis (Figure 3).

Transmission electronic microscopy (TEM) was used to further study the morphology of the 9-methylfascaplysin-modified Aβ sample. The unmodified Aβ oligomer was almost round, and the diameter was about 10–100 nm, which is consistent with previous reports [15]. Interestingly, when Aβ was co-incubated with 9-methylfascaplysin, the resulting oligomers had filiform structures, which is different from that of the typical Aβ oligomer (Figure 4).

### 2.3. Aβ Oligomer-Induced Neurotoxicity in SH-SY5Y Cells is Reduced by Nanomolar 9-Methylfascaplysin

Aβ oligomer can bind to the postsynaptic membrane of neurons, causing neuronal death at low concentrations [16]. To evaluate whether 9-methylfascaplysin could protect against neurotoxicity, SH-SY5Y cells were used. Aβ oligomers were added to SH-SY5Y cells alone, or with 9-methylfascaplysin (1–100 nM), and incubated for 24 h. Aβ oligomers at 1.5 μM killed about 50% of cells, as evidenced by the MTT assay (Figure 5). However, the viability of cells treated with 1 nM 9-methylfascaplysin, together with Aβ, reached about 60%. Moreover, the cellular protective effects of 9-methylfascaplysin increased with the application of this compound at higher concentration.

FDA/PI staining is another method that is typically used to assess the cell protective effects of small molecules such as 9-methylfascaplysin. Using this method, living cells and dead cells are respectively stained green and red. The portion of viable cells in the group treated with Aβ oligomer was less than that in the control group (Figure 6). Furthermore, addition of 100 nM 9-methylfascaplysin significantly reduced Aβ oligomer-induced neurotoxicity in the SH-SY5Y cells.

### 2.4. Suggested Interactions between Aβ_42_ and 9-Methylfascaplysin were Revealed by MD Simulations

MD simulations were performed to investigate the possible interactions between Aβ and 9-methylfascaplysin. The binding free energies of the Aβ-9-methylfascaplysin complex in the final 95–100 ns were calculated. The parent molecule, fascaplysin, was used as a control for comparison. As shown in Table 1, both the polar and nonpolar energy components are favorable for the formation of the Aβ-9-methylfascaplysin complex. Moreover, the polar energy was larger than the nonpolar part, indicating that 9-methylfascaplysin might interact with acidic and basic residues at the *N*-terminus of Aβ_42_. It is noted that the absolute values of the binding free energy of 9-methylfascaplysin is larger than that of fascaplysin. Thus, 9-methylfascaplysin is suggested to be more effective than its lead compound to decrease Aβ aggregation, which is consistent with previous ThT and TEM results.

The interactions between Aβ_42_ and 9-methylfascaplysin in the final 5 ns were determined, as demonstrated in Figure 7. It was shown that 9-methylfascaplysin molecules preferentially interact with the negatively charged residues of D1, E3, D7, and E11 at the N-terminus of Aβ_42_. This interaction of 9-methylfascaplysin with negatively charged residues of Aβ_42_ mainly contribute to the calculated polar binding energy. Additionally, in the same binding mode, 9-methylfascaplysin molecules can have contact with hydrophobic residues F4, Y10, F20, V39, and I41 of Aβ_42_, which contributes to the determined nonpolar binding energy. However, no atomic contact was observed between the nearby positively charged residues (R5, K16, and K28) and 9-methylfascaplysin molecules, likely due to the fact that 9-methylfascaplysin contains a positively charged quaternary nitrogen atom (Figure 7B). Therefore, 9-methylfascaplysin molecules might be repelled by positively charged residues, and get attracted by negatively charged ones. The number of contacts calculated between 9-methylfascaplysin and negatively charged residues of Aβ_42_ were, expectedly, far more than those between 9-methylfascaplysin and positively charged residues.

The hydrogen bonds that could form between Aβ_42_ and 9-methylfascaplysin were also calculated, as shown in Figure 8. During the period of the simulation, the number of hydrogen bonds formed between Aβ_42_ and 9-methylfascaplysin was small, at around 3–5. This may result from 9-methylfascaplysin only containing one oxygen atom hydrogen bond acceptor and one nitrogen atom hydrogen bond donor. As shown in Figure 8B,C, the amino hydrogen of 9-methylfascaplysin could participate in hydrogen bonding with residues D7 and I41 of Aβ_42_.

The π–π interactions between Aβ_42_ and 9-methylfascaplysin were also calculated, and are shown in Figure 9. It is shown that 9-methylfascaplysin could interact via π–π interactions with the aromatic F4, F20, and Y10 residues of Aβ_42_. Such interactions contributed mainly to the calculated nonpolar energy component. These results were also supported by the observation that the number of contacts between 9-methylfascaplysin and the residues F4, F20, and Y10, were relatively high, suggesting that these residues would preferentially interact with 9-methylfascaplysin.

## 3. Discussion

In this study, it was found, for the first time, that the marine-derived alkaloid, fascaplysin, can inhibit Aβ fibrilization. Moreover, a new analogue, 9-methylfascaplysin, was synthesized and found to be more potent than fascaplysin to inhibit Aβ fibril formation. This molecule, 9-methylfascaplysin, was also found to prevent the formation of Aβ oligomer in vitro. Furthermore, it was discovered that 9-methylfascaplysin protects against the neurotoxicity of Aβ oligomer in SH-SY5Y cells, in vitro, at very low concentrations. Molecular dynamics simulations were conducted, and suggested that interactions between 9-methylfascaplysin and Aβ corresponded with some hydrogen bonding and many hydrophobic interactions, including π–π interactions, which is consistent with other bioactive small compounds including EGCG, hematoxylin, and brazilin [17,18,19]. Moreover, our previous studies have shown that the intermolecular interactions in Aβ aggregates are also dominated by both hydrophobic interactions and electrostatic interactions, including hydrogen bonding [20]. Therefore, the direct interactions between 9-methylfascaplysin and Aβ appear to compete against the intermolecular interactions among Aβ aggregates, thus inhibiting the unfolding and subsequent self-assembly of Aβ.

Fascaplysin is a β-carboline alkaloid that was derived from a marine sponge. The β-carboline structure of fascaplysin was suggested to play an important role in its biological activity. β-Carbolines were first discovered in plants, and tend to possess pharmacological properties related to neurological disorders, such as Parkinson’s disease [21,22]. β-Carbolines have also been reported to have a potential for treating AD. For example, Zhao et al. designed and synthesized a series of novel bivalent β-carbolines and found that some exhibited neuroprotection against Aβ-induced toxicity in neuronal cells [23]. Horton et al. demonstrated that some other β-carbolines have moderate effects on preventing the fibrilization of Aβ [24]. In the present study, it was found that fascaplysin also inhibits Aβ fibrillization, providing further evidence that β-carbolines could be useful to treat AD. Moreover, the fascaplysin derivative, 9-methylfascaplysin, had strong biological activity in inhibiting Aβ aggregation. The synthesis of 9-methylfascaplysin takes place at temperatures above 200 °C, and this molecule is stable at room temperature. Moreover, 9-methylfascaplysin easily dissolves in DMSO at 10 mM and in ddH_2_O at 1 mM. 9-Methylfascaplysin was predicted to be able to penetrate into the brain by ACD/Percepta software. Preliminary pharmacodynamic results have suggested that 9-methylfascaplysin could cross the blood–brain barrier and be retained in the brain, indicating that this molecule might be clinically useful. Therefore, additional experiments were performed using this compound.

Proteins can form amyloid-like fibril structures under certain conditions [25]. The inhibition of Aβ fibril formation is considered to be a possible therapy to reduce neurotoxicity [26]. In the present study, 9-methylfascaplysin was shown to better inhibit the formation of Aβ fibrils than fascaplysin or curcumin. Curcumin has been previously shown to bind with fibrils, and to inhibit Aβ fibril formation, largely attributing to its structural property of having two aromatic end groups and their co-planarity [27]. 9-Methylfascaplysin possesses similar structural characteristics, and may bind with Aβ fibrils in a similar way.

Aβ oligomers are considered to be highly neurotoxic. These oligomers can interact with cellular membranes and are harmful to neurons [28]. 9-Methylfascaplysin can interact directly with Aβ through π–π interactions and hydrogen bonds, proposed from MD simulations, to inhibit the formation of neurotoxic Aβ oligomers. It was also shown here that 9-methylfascaplysin can protect SH-SY5Y cells at nanomolar concentrations. Many other marine-derived active compounds can also inhibit the cytotoxicity of Aβ oligomer but, typically, at higher concentrations. To name just one example, Harms found that the steroids isolated from *Callyspongia* cf. *C. flammea* were able to protect neurons against Aβ, and suggested that marine compounds have the potential to inhibit Aβ aggregation [29]. Compared to literature reports, 9-methylfascaplysin appears to be one of the most potent marine-derived compounds that produces in vitro neuroprotective effects against Aβ aggregation.

As it relates to the future potential of 9-methylfascaplysin to be used as an AD therapeutic, the toxicity of this molecule should be considered. It has been reported that the related natural product, fascaplysin, is cytotoxic to cancer cell lines at micromolar concentrations [30]. In the present study, it was found that the new molecule, 9-methylfascaplysin, protects against the neurotoxicity of Aβ oligomers at nanomolar concentrations. Moreover, 9-methylfascaplysinunder at 0.2 μM appears to be safe in SH-SY5Y cells. Since significantly sub-cytotoxic concentrations of 9-methylfascaplysin provided neuroprotective effects in vitro, this molecule may be safe to use in the treatment of AD. Moreover, other β-carbolines have been suggested to have cytotoxic as well as neuroprotective properties [31]. Therefore, it will be interesting and important to further investigate the safety profile of 9-methylfascaplysin in vivo, and develop new analogues of this compound with reduced toxicity and/or improved neuroprotective activity by chemical structural modification.

Lastly, there are other targets of AD that will be important to consider in the development of fascaplysin analogues. Although not here tested, it could be speculated that 9-methylfascaplysin might inhibit acetylcholinesterase activity and induce P-glycoprotein, as fascaplysin does, which would make fascaplysin derivatives a class of potential multiple-target anti-AD drugs [12].

## 4. Materials and Methods

### 4.1. The Synthesis of Fascaplysin and 9-Methylfascaplysin

Fascaplysin and 9-methylfascaplysin were synthesized via a two-step process from tryptamine and 5-methyltryptamine in 50–55% overall yields, according to published methods, as shown in Figure 1 [32,33,34]. The spectroscopic and spectrometric data of **3a** matched with literature reports [35]. **2b** (0.120 g) was sealed in a stainless-steel hydrothermal reactor under N_2_, and heated to 230 °C for 80 min. The reaction mixture was dissolved in MeOH and filtered through nylon membrane to remove carbonized particles. The filtrate was precipitated in ethyl acetate. The obtained precipitate was dried under vacuum to afford product **3b** in 80% yield. The purity of **3a** and **3b** were determined to be 99.0% by HPLC (Appendix A). **3a**: ^1^H NMR (400 MHz, DMSO-d_6_) δ 9.62 (d, *J =* 6.2 Hz, 1H), 9.14 (d, *J =* 5.9 Hz, 1H), 8.57 (d, *J =* 8.0 Hz, 1H), 8.49 (d, *J =* 8.1 Hz, 1H), 8.07 (d, *J =* 7.5 Hz, 1H), 8.01 (d, *J =* 7.9 Hz, 1H), 7.89 (t, *J =* 7.8 Hz, 1H), 7.82 (d, *J =* 8.3 Hz, 1H), 7.74 (t, *J =* 7.5 Hz, 1H), 7.53 (t, *J =* 7.5 Hz, 1H). ^13^C NMR (101 MHz, DMSO-d_6_): δ 182.83, 147.63, 147.57, 141.01, 137.50, 134.77, 131.80, 131.66, 127.20, 126.04, 124.84, 124.53, 123.48, 123.31, 120.79, 119.96, 116.09, 114.27. HRMS (ESI) (positive mode): *m*/*z* calculated for C_18_H_11_N_2_O: 271.0866; M^+^, found: 271.0864.

**3b**: dark brown solid, mp > 300 °C. ^1^H NMR (400 MHz, methanol-d_4_): δ 9.32 (s, 1H), 8.85 (s, 1H), 8.33 (d, *J =* 8.1 Hz, 1H), 8.19 (s, 1H), 8.06–7.96 (m, 2H), 7.75 (t, *J =* 7.5 Hz, 1H), 7.69–7.57 (m, 2H), 2.51 (s, 3H). ^1^H NMR (400 MHz, DMSO-d_6_): δ 13.36 (s, 1H), 9.62 (d, *J =* 6.2 Hz, 1H), 9.08 (d, *J =* 6.2 Hz, 1H), 8.49 (d, *J =* 8.1 Hz, 1H), 8.33 (s, 1H), 8.09–7.97 (m, 2H), 7.73 (t, *J =* 7.5 Hz, 1H), 7.73–7.62 (m, 2H). ^13^C NMR (101 MHz, DMSO-d_6_): δ 182.74, 147.56, 145.79, 140.51, 137.49, 136.39, 132.86, 131.73, 131.46, 127.08, 126.01, 124.48, 123.95, 123.09, 120.62, 120.05, 116.08, 113.88, 21.37. HRMS (ESI) (positive mode): *m*/*z* calculated for C_19_H_13_N_2_O: 285.1022; M^+^, found: 285.1014.

### 4.2. Preparation of Aβ Oligomer

Aβ oligomer was prepared as previously reported [15]. Synthetic Aβ_42_ (GL Biochem, Shanghai, China) was dissolved in hexafluoroisopropanol (HFIP, Sigma, St. Louis, MO, USA). Furthermore, 100 μL Aβ was added to a clean tube and diluted with 900 μL Milli-Q water. HFIP in the solution was evaporated under N_2_ to until a final concentration of Aβ of about 50 μM. A portion of Aβ was mixed with different concentrations of 9-methylfascaplysin, or a vehicle blank, to obtain an Aβ solution at 10 μM. The Aβ solution was kept for two days at room temperature under constant stirring before it was centrifuged at 14,000× *g* for 15 min at 4 °C. The supernatant was collected, and this mainly contained Aβ oligomer.

### 4.3. Preparation of Aβ Fibril and Thioflavin-T (ThT) Assay

Aβ fibrils were also prepared as previously described [36]. Aβ_42_ power was dissolved in HFIP to form Aβ monomers and diluted with Milli-Q water. After completely evaporating the HFIP with N_2_, Aβ monomer was diluted with NaOH to form 1 mM Aβ solution. Next, 2 μL Aβ solution and 10 μL 9-methylfascaplysin was added to PBS that contained 5 μM ThT (Sigma, St. Louis, MO, USA). The mixture (200 μL) was incubated in the dark for 6 days at 37 °C. Fluorescence intensity at the excitation and emission wavelengths of 440 nm and 485 nm, respectively, was used to evaluate the quantity of Aβ fibrils.

### 4.4. Dot Blotting Analysis

Dot blotting analysis was also performed according to literature methods. Briefly, 2 μL co-incubated Aβ and 9-methylfascaplysin was spotted onto the nitrocellulose membrane before it was dried by air. After blocking the membrane in 1% TBST solution containing 5% BSA overnight, the membrane was incubated with anti-oligomer antibody A11 (Thermo Fisher Scientific, Waltham, MA, USA, 1:1000) or anti-Aβ1-17 antibody 6E10 (Sigma, St. Louis, MO, USA, 1:1000) for 1 h with shaking. Subsequently, the membrane was washed by 1% TBST before it was re-incubated with antibody for 1 h. The membrane was finally developed with an Enhanced chemiluminescence plus kit after 3 washes in 1% TBST.

### 4.5. TEM Analysis

TEM samples were performed by placing 2 μL samples on a carbon-coated grid. The samples were stained by 1% uranyl acetate, and the excess staining solution was removed by placing the grid onto a clean paper. The grid was examined using a TEM (JEOL, Tokyo, Japan).

### 4.6. SH-SY5Y Cell Culture

SH-SY5Y cells (Chinese Academy of Sciences, Shanghai, China) were maintained in high glucose modified Eagle’s medium (DMEM) which was supplemented with 1% penicillin (100 U/mL)/streptomycin (100 µg/mL) and 10% fetal bovine serum (FBS) with 5% CO_2_ at 37 °C. The medium was refreshed every two days. Before experiments, the medium of SH-SY5Y was changed to DMEM supplemented with 1% FBS, for 24 h.

### 4.7. Cell Viability Measurement

3-(4,5-Dimethylthiazol-2-yl)-2,5diphenyltetrazolium bromide (MTT) assay was performed to measure the cell viability. An aliquot of 5 μL Aβ oligomer or 9-methylfascaplysin-modified oligomer was added to the wells of 96-well plates. After incubating the cells for 24 h at 37 °C, 10 μL MTT was added. Next, 100 µL solvate (0.01 N HCl in 10% SDS) was added after 4 h. The absorbance of the samples was measured at a wavelength of 570 nm with 655 nm after 16 h.

### 4.8. FDA/PI Double Staining

In this protocol, viable cells were stained by fluorescein diacetate (FDA), which form fluorescein in viable cells to be visualized. By contrast, propidium iodide (PI) stained nonviable cells by entering their cell membranes. Briefly, after staining with 5 μg/mL of PI and 10 μg/mL of FDA for 15 min, the cells were examined. The images were captured by a UV light microscopy and compared with photos taken under phase contrast microscopy. The photos of each well were captured from five random fields in order to measure cell viability quantitatively. The number of cells stained by PI and FDA were counted to calculate the cell viability as follows: % of cell viability = the number of cells stained by FDA/(the number of cells stained by PI + number of cells stained by FDA) [37].

### 4.9. Molecular Dynamics Simulations

The initial coordinates for Aβ_42_ monomer used in the simulation were taken from the solution NMR structures (model 1, PDB ID: 1IYT) in Protein Data Bank (PDB) (https://www.rcsb.org/), and the conformation of Aβ_42_ monomer is shown in Figure 7 [38,39]. The structure of 9-methylfascaplysin and its lead compound, fascaplysin, were produced using the software ChemBioDraw (13.0, PerkinElmer, Waltham, MA, USA). The chemical structures of 9-methylfascaplysin are also shown in Figure 7. All MD simulations were performed using the Gromacs 5.1.4 software package with GROMOS96 54A7 force field [40,41]. The GROMOS96 54A7 force field parameters of fascaplysin and 9-methylfascaplysin were obtained from the website of the Automated Topology Builder (ATB) and Repository version 3.0 (https://atb.uq.edu.au/) [42], which has widely been used to obtain the GROMOS force field parameters of chemical compounds [43]. The SPC/E model was used to represent the water. First, Aβ_42_ monomer was put in an 8 × 8 × 8 cubic box with periodic boundary conditions. Then, eight inhibitors were randomly placed around the Aβ_42_ monomer. The minimum distance among any molecule was set to 15 Å. The concentration of 9-methylfascaplysin and fascaplysin used in this system was 26.94 μM. The remaining space was filled with water molecules. Finally, in order to neutralize the simulation system, 5 Cl^−^ ions were added, in random locations, to replace the corresponding water molecules.

It was necessary to minimize the simulation system, and the collision of any atoms in the simulation system and incorrect geometry was removed. The steepest descent minimization method with 50,000 steps was used to allow reasonable spacing among water molecules, 9-methylfascaplysin, and the Aβ_42_ monomer. Short-range van der Waals interactions and short-range electrostatic interactions were smoothly truncated at 10 Å. Newton’s equation of motion was integrated using the leapfrog algorithm with a time step of 2 fs for equilibrium dynamics [44,45]. The equilibrium MD simulations under separate isothermal–isometric and isothermal–isobaric (NPT) ensembles were performed for 100 ps each. Maxwell distribution was used to produce the initial velocity. All covalent bonds involving hydrogen atoms were constrained by the LINCS algorithm [46]. Long-range electrostatic interactions were calculated using the particle mesh Ewald method (PME) with a grid size of 0.16 Å [47]. The final MD simulations were performed for 100 ns under an NPT ensemble. The temperature of 310 K and pressure of 1 atm were controlled using the Parrinello–Rahman and v-rescale method, respectively [48]. For all MD simulations, the atomic coordinates were saved every 50 ps for further data analysis.

Several auxiliary programs in GROMACS 5.1.4 package were used to analyze the simulation trajectories. The gmx hbond command was used to calculate the number of hydrogen bonds between Aβ_42_ monomer and 9-methylfascaplysin molecules. The command of gmx mindist was used to analyze the number of contacts between each residue of Aβ_42_ monomer with 9-methylfascaplysin molecules. The binding energy between inhibitors with the Aβ_42_ monomer was calculated using g_mmpbsa software [49,50]. Temporal snapshots were obtained using Visual Molecular Dynamics (VMD) 1.9.4 (University of Illinois at Urbana–Champaign, Urbana and Champaign, Champaign, IL, USA) [51].

### 4.10. Statistics Analysis

Data were expressed as means ± SEM. Statistical significance was determined by one-way ANOVA and Tukey’s test for post hoc multiple comparisons. Results with *p* < 0.05 were considered to be statistically significant.

## 5. Conclusions

In conclusion, it was demonstrated that 9-methylfascaplysin inhibits the aggregation of toxic Aβ in vitro, possibly directly via hydrogen bonding and π–π interactions. The interaction of 9-methylfascaplysin with Aβ could explain the inhibition of the neuronal toxicity of Aβ oligomers in SH-SY5Y cells. Based on these results, it is suggested that fascaplysin derivatives might be developed as potential new drugs for AD treatment. Further studies, including medicinal chemistry structural optimization and in vivo toxicity, should be conducted using the 9-methylfascaplysin scaffold as a lead molecule.

## Figures and Tables

**Figure 1 marinedrugs-17-00121-f001:**
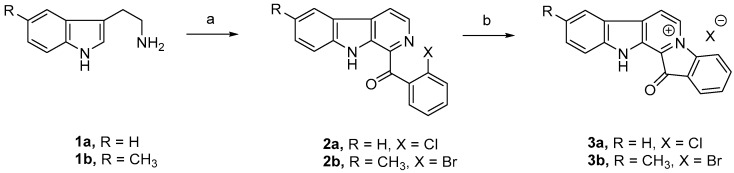
Synthesis of fascaplysin (**3a**) and 9-methylfascaplysin (**3b**). Reaction conditions: (**a**) *o*-halo acetophenone, I_2_/H_2_O_2_, DMSO, reflux; (**b**) 220–230 °C, 20–80 min.

**Figure 2 marinedrugs-17-00121-f002:**
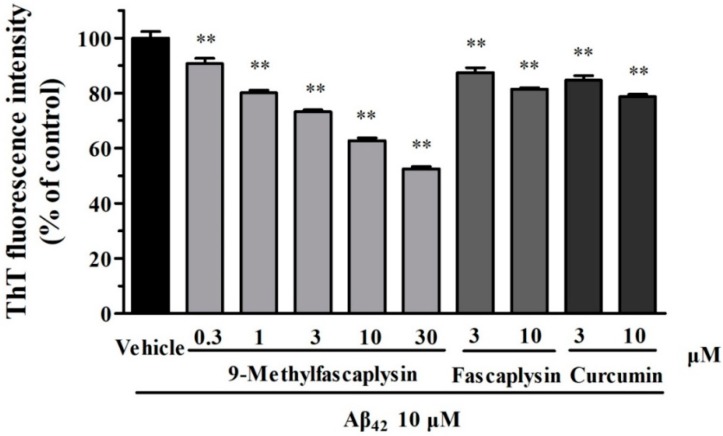
9-Methylfascaplysin can inhibit the formation of Aβ_42_ fibril in a concentration-dependent manner. Aβ_42_ monomers (10 μM) were incubated with different concentrations of 9-methylfascaplysin (0.3–30 μM), fascaplysin (3–10 μM), or curcumin (3–10 μM). The results were analyzed by Thioflavin-T (ThT) assay. The data shown represent the mean ± SEM of three separate experiments; ** *p* < 0.01 versus the control group (one-way ANOVA and Tukey’s test).

**Figure 3 marinedrugs-17-00121-f003:**
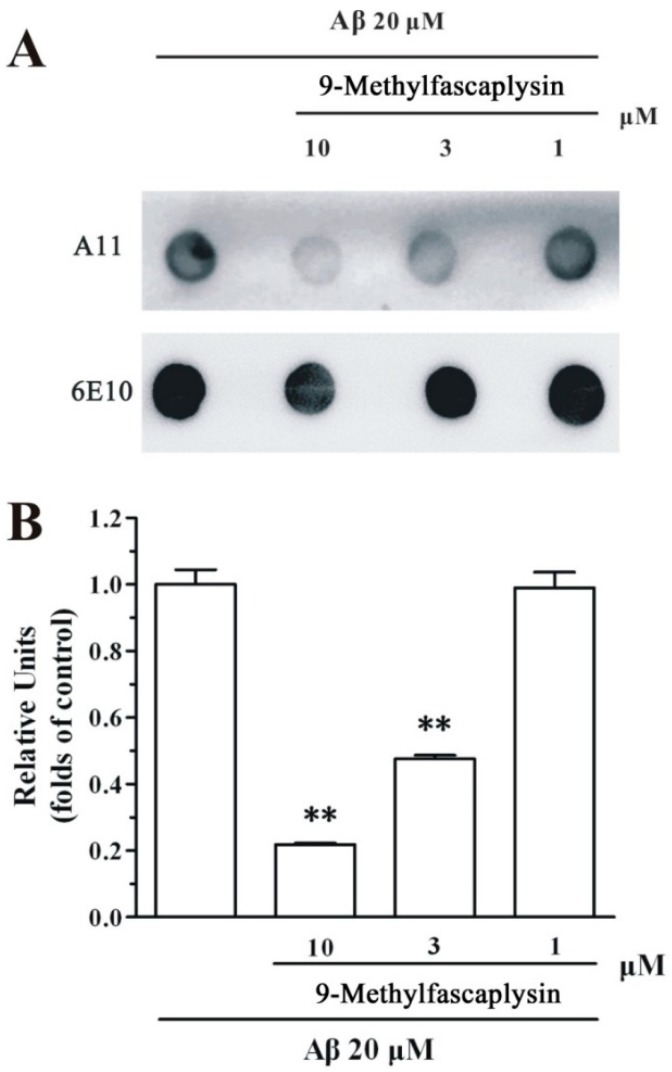
9-Methylfascaplysin inhibits the aggregation of Aβ_42_ oligomer in a concentration-dependent manner. (**A**) A 20 μM Aβ_42_ monomer solutions, with or without different concentrations of 9-methylfascaplysin (1–10 μM), were continuously vibrated for 48 h. The supernatant was spotted on the membrane after centrifuging the solution. The two membranes were incubated with A11 and 6E10 antibodies, respectively. (**B**) The optical density of dots was quantified by ImageJ. The data shown represent the mean ± SEM, ** *p* < 0.01 versus the control group (one-way ANOVA and Tukey’s test).

**Figure 4 marinedrugs-17-00121-f004:**
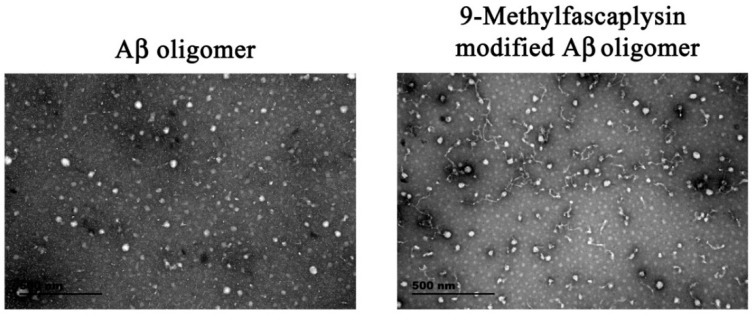
9-Methylfascaplysin can change the morphology of the Aβ_42_ oligomer. Typical Aβ_42_ monomers and 10 μM 9-methylfascaplysin-modified Aβ_42_ monomers were separately incubated for two days to form oligomers. After centrifuging for 15 min, the supernatant was observed by TEM.

**Figure 5 marinedrugs-17-00121-f005:**
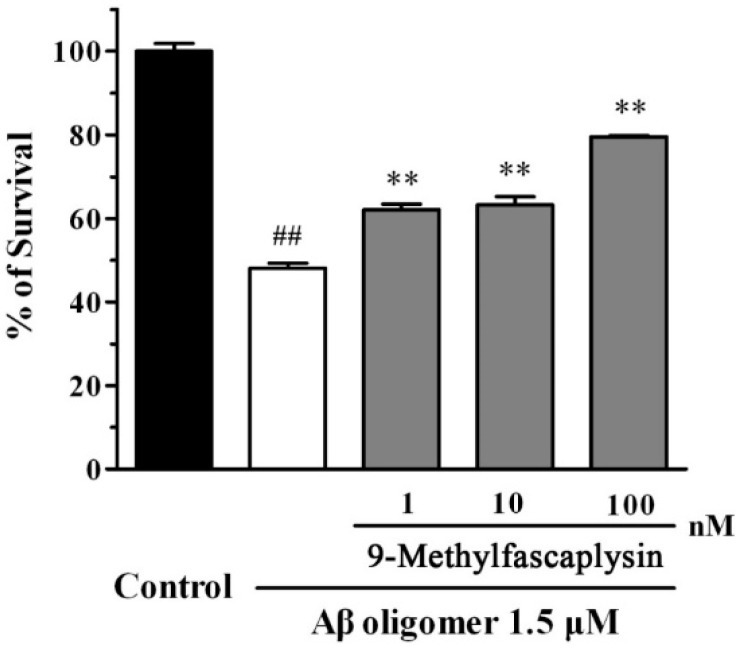
9-Methylfascaplysin can protect against the toxic of Aβ_42_ oligomers in SH-SY5Y cells. High concentration Aβ_42_ monomer was diluted to 1.5 μM with Milli-Q water or different concentrations of 9-methylfascaplysin (1–100 nM). The solutions were vibrated for 24 h, and samples were further added to the wells in 96-well plates. After incubating for 24 h, the MTT assay was used to analyze cell viability. Data reported are the mean ± SEM, ^##^
*p* < 0.01 versus the control group, ** *p* < 0.01 versus the Aβ oligomer group (one-way ANOVA and Tukey’s test).

**Figure 6 marinedrugs-17-00121-f006:**
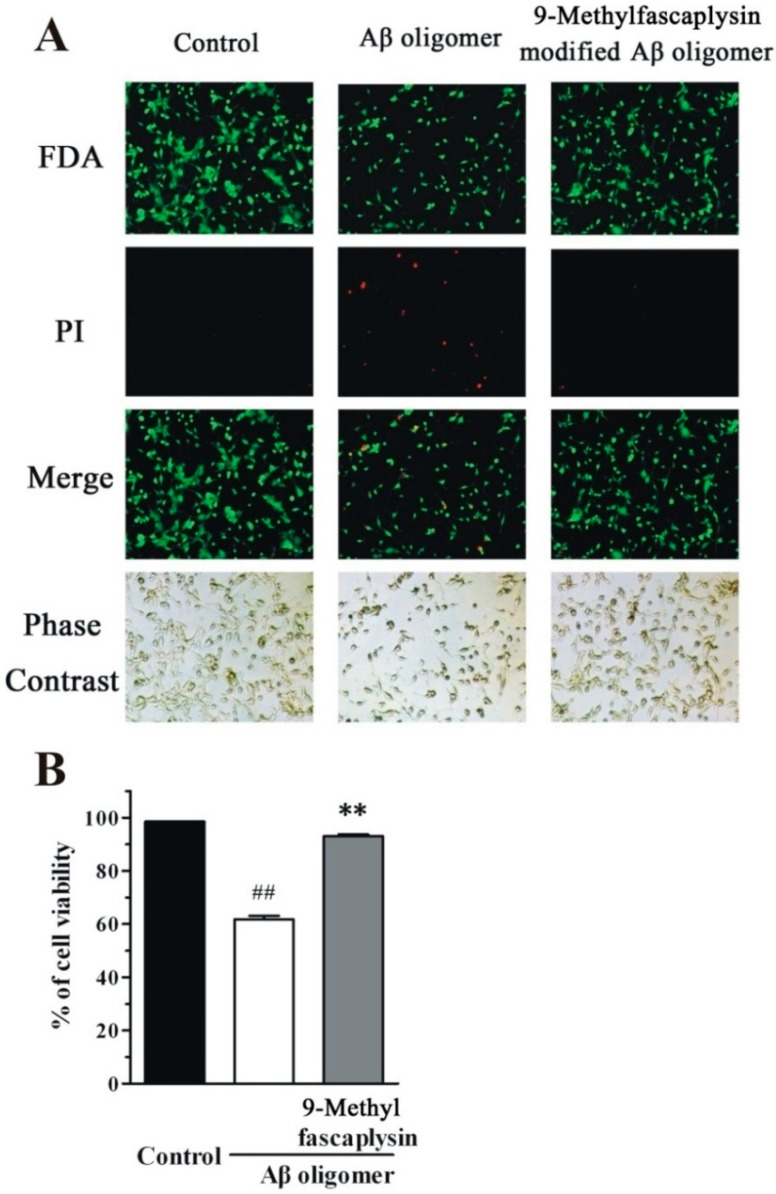
Nanomolar 9-methylfascaplysin reduces Aβ_42_-induced neurotoxicity in SH-SY5Y cells. (**A**) Aβ_42_ oligomer and 100 nM 9-methylfascaplysin-modified Aβ oligomer were added to 6-well plates. FDA/PI double staining was used to demonstrate cell viability. (**B**) Quantitative analysis of the FDA/PI double staining. The results represent the mean ± SEM of three separate experiments; ^##^
*p* < 0.01 versus the control group, ** *p* < 0.01 versus the Aβ oligomer group (one-way ANOVA and Tukey’s test).

**Figure 7 marinedrugs-17-00121-f007:**
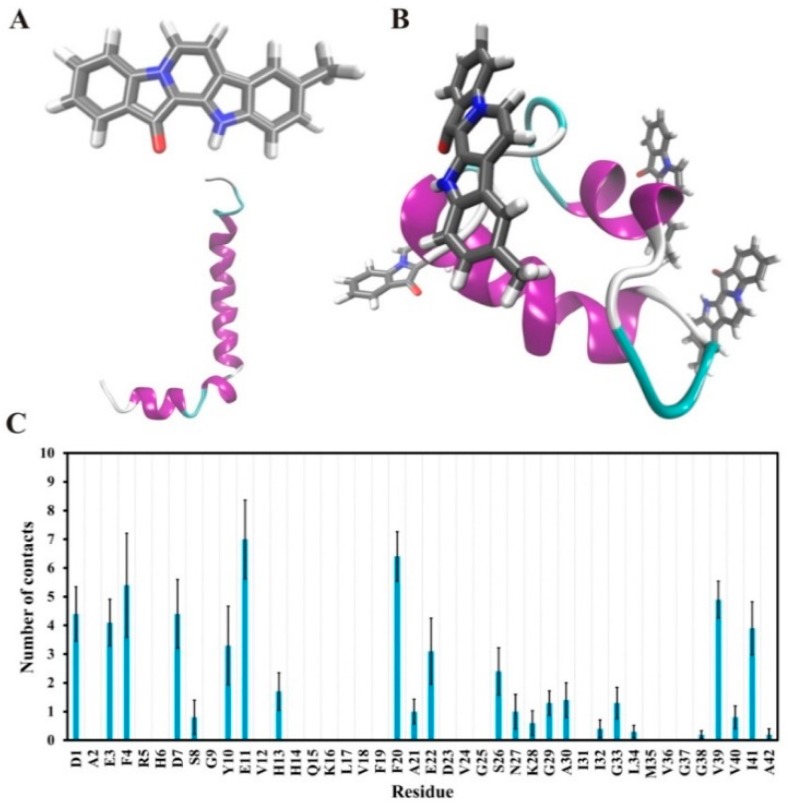
Suggested interactions between Aβ_42_ and 9-methylfascaplysin in MD simulations. (**A**) The initial structures of 9-methylfascaplysin (upper) and Aβ_42_ monomer (lower). (**B**) The typical conformation of the Aβ–9-methylfascaplysin complex. Aβ_42_ is displayed as a new cartoon model, with the purple region representing the α-helix. 9-Methylfascaplysin is represented using a licorice model. (**C**) The contact numbers of each residue of 9-methylfascaplysin with Aβ_42_ as calculated using the MD trajectories of the final 95–100 ns.

**Figure 8 marinedrugs-17-00121-f008:**
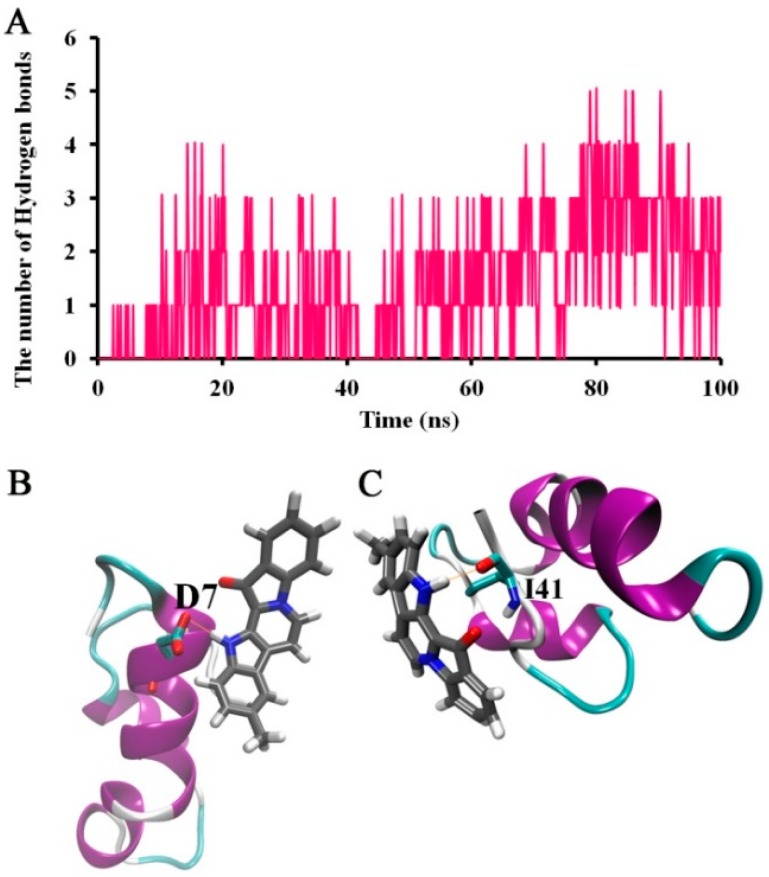
Suggested hydrogen bonding interactions between Aβ_42_ and 9-methylfascaplysin. (**A**)Time-dependent plot of the number of the hydrogen bonds between Aβ_42_ and 9-methylfascaplysin. (**B**,**C**) The typical snapshot of hydrogen bonding between Aβ_42_ and 9-methylfascaplysin. The hydrogen bonding is represented by an orange dotted line.

**Figure 9 marinedrugs-17-00121-f009:**
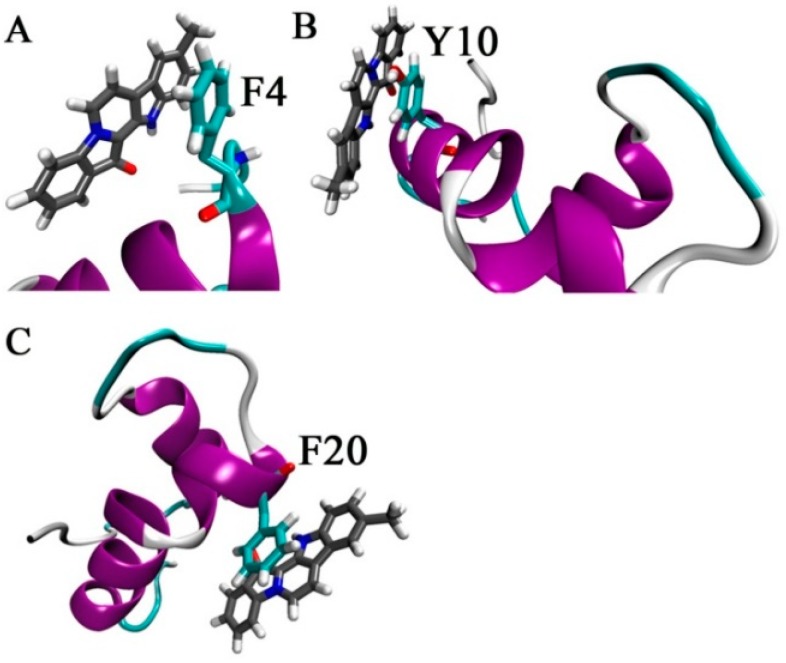
Typical temporal snapshots in the simulation shows the π–π interactions suggested to occur between Aβ_42_ and 9-methylfascaplysin. The aromatic residues (**A**) F4, (**B**)Y10, and (**C**) F20 interact with 9-methylfascaplysin via π–π interactions.

**Table 1 marinedrugs-17-00121-t001:** Components of binding free energy of Aβ and fascaplysin and its variant 9-methylfascaplysin.

Energy Components	Aβ_42_–9-Methylfascaplysin Complex (kJ/mol)	Aβ_42_–Fascaplysin Complex (kJ/mol)
van der Waals	−211.88	−176.05
Electrostatic	−730.19	−829.99
Polar solvation	365.30	522.74
SASA	−27.13	−23.06
Polar	−364.89	−307.25
Nonpolar	−239.01	−199.11
Binding	−603.90	−506.36

Polar = Electrostatic + Polar solvation, Nonpolar = van der Waals + SASA, Binding = Polar + Nonpolar. SASA: solvent-accessible surface area.

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
