# Peer review of "9-Methylfascaplysin Is a More Potent Aβ Aggregation Inhibitor than the Marine-Derived Alkaloid, Fascaplysin, and Produces Nanomolar Neuroprotective Effects in SH-SY5Y Cells"

_marinedrugs, 2019, doi:10.3390/md17020121_

Round 1
Reviewer 1 Report
The manuscript "9-Methylfascaplysin is A More Potent Aβ Aggregation Inhibitor than the Marine-Derived Alkaloid, Fascaplysin, and Produces Nanomolar Neuroprotective Effects in SH-SY5Y Cells" by Sun et al. have synthesized two compounds with potential application for treatment of Alzheimer’s disease. They have also presented molecular mechanism and in-silico studies, but the experimental section is not performed in sufficient details.
The author synthesized 9-Methylfascaplysin which is better in activity. It is advisable to include some background information for precise synthesis of effective compound than previous molecule. If they synthesized many molecules, then appropriate information is highly commended.
Fascaplysin rapidly decreases cell viability (doi:10.3390/ijms18102074) with promising cytotoxicity 2-3 micromolar but the activity range has been assessed in higher concentration such as 1,3 and 10 micromolar (Fig 3).Hence the authors should also assess such cytotoxic effect in the cell-line in Figure 5 or information about IC50 is highly recommended. It would be highly interesting if the compound has lower cytotoxicity while retaining higher biological potency.
The molecular targets involved in Aβ oligomer formation need to be explained and the impact of compounds on such molecular targets should be assessed by western-blot analysis.
The author should also perform MD studies with fascaplysin and explain why 9-Methylfascaplysin is better drug candidate than fascaplysin.
It is advisable to authors for performing the analysis of physiochemical properties as stability, solubility etc.
Author Response
Please find the response in the attached file.

Reviewer 2 Report
Manuscript from Sun group describe synthesis and antiAβ activity of new 9-methylfascaplysin compound which could be a potential therapeutic agent for Alzheimer’s Disease (AD). Author reported synthesis and various activity such as dot blotting, TEM, and neuroprotective assay in SH-SY5Y cell lines. Furthermore MD simulations were performed to understand interactions of Aβ and 9-methylfascaplysin which seems supportive of their funding with hydrogen bonding and π-π interactions. Overall a very interesting studies performed with this new analogue of fascaplysin. Manuscript is clearly written and provided full details of experiments and data were carefully analyzed. Manuscript can be accepted after minor revision for the comments below.
1. Author should perform some stability and blood-brain barrier permeation studies for 9-methylfascaplysin so that this analog could be clinically useful.
2. Why author only synthesized Methyl analogues?
3. Provide purity evidence for 9-methylfascaplysin.
Author Response

(The authors gave the same response as above.)

Round 2
Reviewer 1 Report
In the revised form of manuscript "9-Methylfascaplysin is a more potent Aβ aggregation inhibitor than the marine-derived alkaloid, fascaplysin, and produces nanomolar neuroprotective effects in SH-SY5Y cells", the authors have fully addressed the major issues by performing additional experiments and providing the rational discussions. Hence, I would like to recommend the article for publication in current form.